# Effect of the COVID-19 Pandemic on Lower Respiratory Tract Infection Determinants in Thai Hospitalized Children: National Data Analysis 2015–2020

**DOI:** 10.3390/tropicalmed7080151

**Published:** 2022-07-28

**Authors:** Rattapon Uppala, Phanthila Sitthikarnkha, Sirapoom Niamsanit, Sumitr Sutra, Kaewjai Thepsuthammarat, Leelawadee Techasatian, Nattachai Anantasit, Jamaree Teeratakulpisarn

**Affiliations:** 1Department of Pediatrics, Faculty of Medicine, Khon Kaen University, 123 Mittraphap Road, Muang, Khon Kaen 40002, Thailand; sirani@kku.ac.th (S.N.); sumitr@kku.ac.th (S.S.); leelawadee@kku.ac.th (L.T.); jamtee@kku.ac.th (J.T.); 2Clinical Epidemiology Unit, Faculty of Medicine, Khon Kaen University, Khon Kaen 40002, Thailand; ckaewj@kku.ac.th; 3Department of Pediatrics, Ramathibodi Hospital, Mahidol University, Bangkok 10400, Thailand; nattachai.ant@mahidol.ac.th

**Keywords:** lower respiratory tract infection, COVID-19, children, hospitalization, nonpharmaceutical interventions

## Abstract

Background: The COVID-19 outbreak emerged in January 2020 and remains present in 2022. During this period, nonpharmaceutical interventions (NPIs) have been used to reduce the spread of COVID-19 infection. Nationwide data analysis should be pushed as the new standard to demonstrate the impact of COVID-19 infection on other respiratory illnesses and the reliability of NPIs during treatment. Objective: This study aims to identify and compare the incidence of lower respiratory tract infections (LRTIs) among children in Thailand before and after the emergence of COVID-19. Methods: A retrospective study was carried out in hospitalized children under the age of 18 in Thailand from October 2015 to September 2020. The International Statistical Classification of Diseases and Related Health Problems, 10th Revision, Thai Modification, was used to identify patient diagnoses (ICD-10-TM). The data were extracted from the Universal Coverage Health Security Scheme Database. Results: A total of 1,610,160 admissions were attributed to LRTIs. The most common diagnosis was pneumonia (61.9%). Compared to the 2019 fiscal year, the number of hospitalizations due to LRTIs decreased by 33.9% in the 2020 fiscal year (COVID-19 period) (282,590 vs. 186,651). The incidence of all three diagnostic groupings was substantially lower in the pre- and post-COVID-19 eras, with a decrease of 28% in the pneumonia group (incidence rate ratio (IRR) = 0.72; 95% confidence interval (CI): 0.71 to 0.72), 44% in the bronchiolitis group (IRR = 0.56; 95% CI: 0.55 to 0.57), and 34% in the bronchitis group (IRR = 0.66; 95% CI: 0.65 to 0.67). Between fiscal years 2019 and 2020, the overall monthly cost of all hospitalizations for LRTIs decreased considerably (*p* value < 0.001). Conclusions: NPIs may decrease the number of pediatric hospitalizations related to LRTIs. All policies designed to prevent the spread of COVID-19 must be continually utilized to maintain the prevention of LRTIs.

## 1. Introduction

Lower respiratory tract infections (LRTIs) represent a major reason for hospitalization, particularly among children under the age of five [1]. In developing countries, this rate has reached 0.22 times per person-year [2]. LRTIs have an effect on health globally due to the high mortality and hospitalization costs of pediatric patients. In children under the age of five, pneumonia accounted for 15% of all causes of death [3]. The majority of LRTIs in children are caused by viruses that regularly spread in schools and regions with dense populations [4]. The peak incidence of LRTIs in children is typically tied to the season and the start of school. In Thailand, LRTIs are a prominent reason why children were hospitalized [1], particularly influenza and respiratory syncytial virus (RSV), which mainly occur from May to October [5].

Infection with SARS-CoV-2 was first detected in China in December 2019 [6]. This virus can cause COVID-19. In March 2020, the World Health Organization (WHO) labeled it a pandemic. COVID-19 displayed multiple clinical symptoms. This condition mainly affects individuals with respiratory illness [7], and primarily spreads by saliva or nasal discharge. To slow its transmission, the WHO advised nonpharmaceutical interventions (NPIs), such as personal protective, environmental, physical distancing, and travel-related measures [8]. Personal protective measures include cleaning hands, wearing masks, and refraining from touching the face. These therapies have been regarded as effective COVID-19 containment measures [9].

In Thailand, the first case of COVID-19 was confirmed in January 2020. This virus’ propagation began around the middle of March 2020. Since March 2020, the Thai government has proposed a number of preventative tactics and policies, such as mask use and isolation. The lockdown began on 3 April 2020 and lasted until May 2020 [10]. The majority of young people remained at home during this time period. After the transmission of COVID-19 at the end of May 2020, the government loosened travel restrictions in June 2020, although overseas travelers still had limitations. Students were required to wear masks at all times, routinely wash their hands, and maintain a safe distance from others.

Since December 2019, the COVID-19 outbreak has not only had a devastating effect on public health, but also affected social and economic expenses. It has caused significant changes in human behavior, including social isolation, working from home, school and childcare facility closures, strict hygiene procedures, and widespread usage of face masks. It has been established that preventative hygiene activities such as hand washing can reduce the transmission of respiratory viruses. The majority of respiratory viruses are spread between individuals by droplet transmission [11]. Therefore, COVID-19 prevention strategies and alterations in human behavior may reduce the transmission of other respiratory viruses, which are a common cause of LRTIs in children [12]. However, the impact of COVID-19 preventive initiatives on childhood LRTIs is still uncertain. We expected NPIs to be able to minimize LRTIs, despite the fact that there is considerable doubt that the NPIs deployed during the outbreak reduced the incidence of certain LRTIs. Therefore, this study investigates the number of LRTIs in the years preceding and following the COVID pandemic. Our study aimed to characterize the number of hospital admissions due to LRTIs in Thai children throughout the fiscal years 2015 and 2020 and to assess the influence of nonpharmacological management during the COVID-19 era on the number of admissions and hospital costs related to LRTIs.

## 2. Materials and Methods

A retrospective study was carried out among Thai children hospitalized due to LRTIs under 18 years of age from October 2015 to September 2020. The periods within this study were divided according to Thailand’s fiscal year, which is from 1 October to 30 September each year. The data were extracted from the National Health Security Office database based on the Universal Coverage Health Security Scheme, which is the main healthcare service in Thailand. Patient diagnoses were identified using the International Statistical Classification of Diseases and Related Health Problems, 10th Revision, Thai Modification (ICD-10-TM) [13]. The LRTIs were defined using ICD-10-TM for acute lower respiratory infection (J09–J21): pneumonia (J09–J18), bronchiolitis (J21), and bronchitis (J20).

We collected data on patients’ age, gender, month and year of admission, hospital level, hospital region, and hospital costs. Hospital level in Thailand was classified into four levels: primary care, secondary care, tertiary care, and private hospital. Primary care is typically located in rural areas, secondary care is typically located far from major cities and offers limited specialization, while tertiary care and private hospitals represent the highest degree of specialty treatment. The institutional review board of Khon Kaen University approved this study (#HE641151).

All statistical analyses were performed using STATA software version 10 (StataCorp LP). Categorical data were described using frequencies and percentages. Continuous data are expressed as the mean and standard deviation. We presented the number of admissions due to LRTIs each year as monthly trends. As COVID-19 measures began in Thailand in the middle of March 2020, the dataset of admission numbers for the 2020 fiscal year (October 2019 to September 2020) was used to compare with the 2019 fiscal year (October 2018 to September 2019). Hence, the admission numbers from March to September 2020 represent post-COVID-19 data. The generalized estimating equation (GEE) was used to produce regression analysis of the number of admissions compared to the monthly trend of admission between the fiscal years 2019 and 2020. Because the number of admissions was the repeated measure variable (some patients may be hospitalized due to LRTIs many times during the study). The incidence of LRTIs was presented per 1000 person-years and divided into pneumonia, bronchiolitis, and bronchitis [1]. We used the incidence rate ratio to compare the incidence rate of each diagnosis between 2015–2019 and 2020. The 95% confidence interval (CI) of the rate was computed based on the normal approximation to the binomial distribution. *p* < 0.05 was considered to indicate statistical significance.

## 3. Results

From the 2015 fiscal year through to the 2020 fiscal year, LRTIs resulted in a total of 1,610,160 admissions among children, predominantly under the age of five years (57.95%). Males were dominant (58.66%). The most common primary diagnosis was pneumonia (61.90%). Thailand’s northeast area had the greatest rate of hospitalization, accounting for 36.81% of all admissions. The majority of children were admitted to secondary-level hospitals (74.48%). We discovered that, when divided by fiscal year, the ratios of age, gender, primary diagnosis, region, and hospital level of admission were comparable from the 2015 fiscal year to the 2020 fiscal year (Table 1). Despite this, the number of admissions in 2020 was 33.94% less than that in 2019 (186,651 vs. 282,590).

From the beginning of the 2015 fiscal year to the end of fiscal year 2019, hospitalizations for LRTIs peaked in September and reached their lowest point in April and May (Figure 1). Admissions declined in fiscal year 2020 compared to the same months in fiscal year 2019. The COVID-19 outbreak in Thailand began around the middle of March 2020. We observed that admissions in April 2020 were considerably lower than those in April 2019 (*p* value < 0.001). After the continuation of the rainy season from May to September 2020, admissions increased but remained lower than in the corresponding month of fiscal year 2019 (*p* value < 0.001) (Figure 2).

During the COVID-19 outbreak in Thailand in the 2020 fiscal year, a total of 186,651 pediatric patients were hospitalized with lower respiratory tract infections (LRTIs). There were 120,281 pneumonia admissions (64.44%), followed by 50,333 bronchitis admissions (26.97%) and 16,037 bronchiolitis admissions (8.59%). Comparing the data to the preceding five years as a benchmark, in the 2020 fiscal year, the incidence of lower respiratory tract infections among pediatric patients decreased statistically significantly (*p* value < 0.001; Table 2) throughout the COVID-19 period compared to the previous fiscal years. The incidence rates of pneumonia, bronchitis, and bronchiolitis were 9.16, 3.84, and 1.22 per 1000 people, respectively. The incidences were lower pre- and post-COVID-19 in all three diagnostic groups, with a 28% reduction in the pneumonia group (incidence rate ratio (RR) = 0.72; 95% confidence interval (CI): 0.71 to 0.72), a 44% reduction in the bronchiolitis group (IRR = 0.56; 95% CI: 0.55 to 0.57), and a 34% reduction in the bronchitis group (IRR = 0.66; 95% CI: 0.65 to 0.67) (Table 3). As depicted in Figure 3, the total monthly cost of all hospitalizations for LRTIs decreases significantly from fiscal year 2019 to fiscal year 2020.

## 4. Discussion

Our data were gathered from the Universal Coverage Health Security Scheme Database, Thailand’s largest database of people, and the number of admissions due to LRTIs among children under the age of 18 was indicated [14]. Gender, age, primary diagnosis, area, and level of hospital admission were the same from fiscal years 2015 to 2020. We found that the number of admissions was highest in fiscal year 2018 and decreased in fiscal year 2020. An earlier study found that influenza infection in China declined during the COVID-19 pandemic [15]. Another study found that pandemic viruses quickly replaced seasonal community-acquired respiratory viruses within three weeks, with pandemic viruses affecting 48% of all identified respiratory viruses. This finding was supported by the fact that pandemic viruses spread more quickly than seasonal viruses [16]. The results were comparable to those found in studies carried out in adults and children in other countries [17,18].

Thailand is a tropical country where respiratory illnesses in children are most prevalent during the rainy season. A previous study in Thailand determined that the majority of admissions occurred between July and October [19]. Thailand’s nationwide lockdown began in the summer of 2020 and continued until the rainy season. After April 2020, along with Thailand’s nationwide lockdown in response to the emergence of COVID-19 clusters, the number of admissions was statistically significantly and drastically reduced. After the COVID-19 epidemic, the number of pediatric outpatient clinics with respiratory tract infections fell concurrently with prior research in China [4]. After implementing community mitigation measures in response to the COVID-19 pandemic, prospective multicenter research conducted in seven U.S. communities revealed a drop in cases of acute respiratory tract infections in children [20]. During the statewide lockdown in Finland in 2020, the daily median rate of pediatric ER visits reduced considerably compared to the study period preceding the lockdown. Additionally, the influenza season was shorter, and the weekly rate of new admission cases declined more quickly. [21]

After the containment of COVID-19 transmission, the lockdown ended in late May 2020, and the number of children admitted due to LRTIs remained lower than that in the previous year. Previous studies showed that parents were hesitant to bring their children to healthcare facilities during this pandemic out of concern about SARS-CoV-2 exposure. Hence, COVID-19′s influence on health care services may be to blame for the drop in hospitalizations, particularly among pediatric patients [22,23]. To prevent the spread of COVID-19, the Thai Ministry of Health utilized a “D-M-H-T-T” policy. It included social distance, mask wearing, hand washing, testing, and the Thai Chana application. The number of hospitalizations dropped in comparison to the previous year after schools reopened with a DMHTT policy that was strictly implemented during the wet season. This indicates that this approach has the potential to prevent the spread of microorganisms that cause LRTIs in children.

We hypothesize that the key factors that may explain the decrease in pediatric LRTIs are the lockdown at the earliest onset of the pandemic outbreak, which decreased the likelihood of transmission; the fact that the entire population wore facemasks, which better protected them from infection; the extensive screening of people with fever, which reduced the number of infection sources; and the active screening tests for people of close contact, which decreased the rate of spread. On the other hand, there may be other reasons for the lower hospitalization rate of LRTIs, such as people’s fear of going to hospital and health care providers’ efforts to cut down on admissions that are not necessary during the pandemic. These factors all may have contributed to the reduction in pediatric LRTIs. Our findings offer more support for the hypothesis that NPIs play a role in determining whether children require hospitalization due to LRTIs. However, lockdowns should be implemented for as short a duration as possible due to economic and behavioral effects [24]. We hypothesize that maintaining a social distance, using masks, and frequent hand washing can minimize the spread of illnesses that affect the respiratory system. Even if COVID-19 measures are no longer in effect, these methods should still be used in open settings to prevent hospitalizations and minimize the overall cost of this issue.

In addition, a Cochrane study that was recently updated has identified and summarized the evidence for these strategies. Hand hygiene was associated with an 11% relative reduction in respiratory illness when compared with no hand hygiene, and physical distancing was associated with a decrease in the number of people who exhibited symptoms of influenza-like illness. The results showed that wearing a surgical mask did not make a difference in the number of patients who had an influenza-like illness when compared to not wearing a mask [11]. Controlling the spread of COVID-19 was made easier by the government’s decision to put preventative measures into effect. These treatments during the peak season for LRTIs result in a decrease in the rate of hospitalization among children, which is a positive outcome. Children may have high levels of virus-induced nonspecific innate immunity and virus-specific adaptive immunological memory from the past, both of which may contribute to the puzzling lower respiratory tract infection rate that is observed in children who are repeatedly exposed to LRTIs. This could be another explanation for the lower rates of LRTIs observed in children. As a result of the absence of immunity to other viruses and the increased vulnerability to serious infection, it is recommended that, during the upcoming seasons, LRTIs should be watched or placed under active monitoring.

There are some limitations to the study. The main limitation is that these findings cannot be extrapolated to other countries. We gathered information on LRTIs that occurred in Thailand for the purposes of this study. In different nations, the COVID-19 pandemic has had varying degrees of impact. Every country has its own culture, policies, and management techniques that are exclusive to that country. Second, the data that we used came from a retrospective study known as the Universal Coverage Health Security Scheme Database. Furthermore, variations in data criteria could have created selection and reporting bias; thus, we are unable to remark on the mortality rate of LRTI hospitalization because we lack the death rate of LRTIs in our data collection. The final limitation is a lack of organism-specific data, which means that the etiology of LRTIs cannot be addressed. Therefore, we cannot explain why the hospitalization rate in fiscal year 2018 is highest compared to previous fiscal years. This study, on the other hand, sheds light on a major shift in epidemiological trends caused by the present COVID-19 infection prevention strategies. The etiological pathogens of LRTIs may be the focus of future research because some pathogens may have an effect on these NPIs, as discovered in a number of countries in a previous study [25].

## 5. Conclusions

NPIs have the potential to lower the number of hospitalizations and costs incurred by children as a result of LRTIs. To continue to prevent LRTIs in Thai children, the ongoing encouragement of any and all measures that stop the spread of COVID-19 infection is needed.

## Figures and Tables

**Figure 1 tropicalmed-07-00151-f001:**
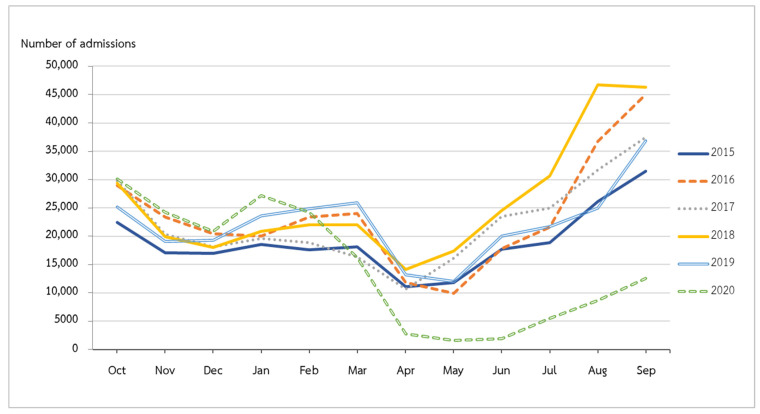
Number of hospitalized LRTIs in fiscal years 2015–2020.

**Figure 2 tropicalmed-07-00151-f002:**
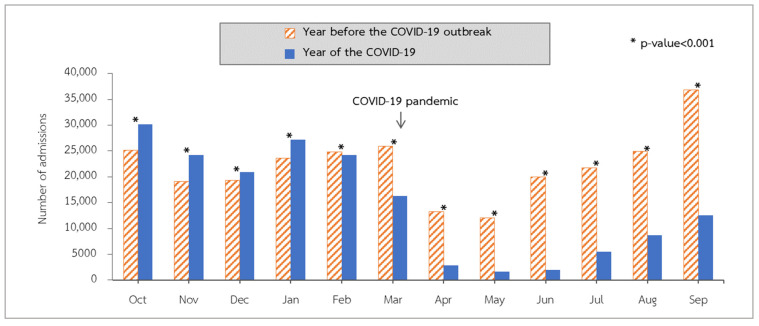
Number of LRTI admissions by month in fiscal year 2019 compared with 2020 in Thailand.

**Figure 3 tropicalmed-07-00151-f003:**
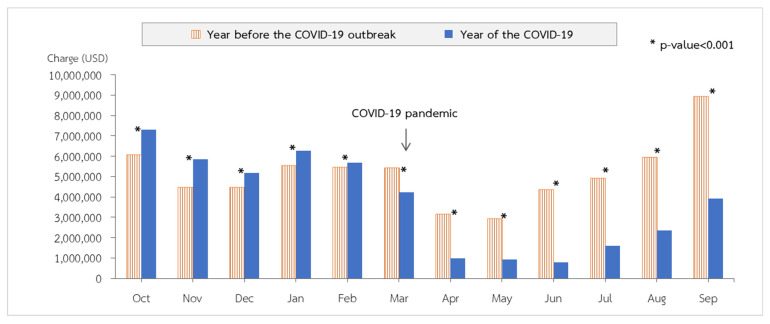
Total costs of all lower respiratory infections by month in fiscal year 2019 compared with 2020 in Thailand.

**Table 1 tropicalmed-07-00151-t001:** The proportion of LRTIs in hospitalized children during the fiscal years 2015–2020.

Fiscal Year	2015	2016	2017	2018	2019	2020	Total
Total	235,316	294,436	281,510	329,657	282,590	186,651	1,610,160
Gender, N (%)							
Male	140,165 (59.56)	171,817 (58.35)	164,619 (58.48)	193,533 (58.71)	164,987 (58.38)	109,400 (58.61)	944,521
Age group, N (%)							
<1 year	57,903 (24.61)	70,313 (23.88)	64,753 (23.00)	82,424 (25.00)	61,280 (21.69)	42,247 (22.63)	378,920
1–<5 years	141,875 (60.29)	174,129 (59.14)	162,349 (57.67)	194,000 (58.85)	157,037 (55.57)	103,718 (55.57)	933,108
5–<18 years	35,538 (15.10)	49,994 (16.98)	54,408 (19.33)	53,233 (16.15)	64,273 (22.74)	40,686 (21.80)	298,132
Principal diagnosis, N (%)							
Pneumonia (J09–J18)	130,943 (55.65)	179,434 (60.94)	176,352 (62.65)	207,382 (62.91)	182,446 (64.56)	120,281 (64.44)	996,838
Bronchiolitis (J21)	31,121 (13.23)	31,535 (10.71)	27,493 (9.77)	34,037 (10.32)	25,067 (8.87)	16,037 (8.59)	165,290
Bronchitis (J20)	73,252 (31.13)	83,467 (28.35)	77,665 (27.59)	88,238 (26.77)	75,077 (26.57)	50,333 (26.97)	448,032
Region, N (%)							
Bangkok	10,304 (4.38)	13,224 (4.49)	11,839 (4.21)	12,464 (3.78)	10,736 (3.80)	6371 (3.41)	64,938
Central	45,402 (19.29)	58,494 (19.87)	55,588 (19.75)	64,445 (19.55)	55,122 (19.51)	34,669 (18.57)	313,720
East	13,827 (5.88)	16,947 (5.76)	16,609 (5.90)	18,719 (5.68)	15,492 (5.48)	8922 (4.78)	90,516
Northeast	83,335 (35.41)	111,398 (37.83)	105,592 (37.51)	125,414 (38.04)	98,332 (34.80)	68,684 (36.80)	592,755
North	20,424 (8.68)	23,906 (8.12)	22,609 (8.03)	28,562 (8.66)	26,815 (9.49)	18,428 (9.87)	140,744
West	12,360 (5.25)	15,735 (5.34)	14,435 (5.13)	18,356 (5.57)	15,730 (5.57)	10,162 (5.44)	86,778
South	49,664 (21.11)	54,732 (18.59)	54,838 (19.48)	61,697 (18.72)	60,363 (21.36)	39,415 (21.12)	320,709
Level of hospital, N (%)							
Primary	16,299 (6.93)	21,319 (7.24)	20,683 (7.35)	25,926 (7.86)	21,032 (7.44)	14,049 (7.53)	119,308
Secondary	173,335 (73.66)	215,586 (73.22)	208,757 (74.16)	246,822 (74.87)	212,567 (75.22)	142,217 (76.19)	1,199,284
Tertiary	37,898 (16.11)	47,275 (16.06)	43,263 (15.37)	48,288 (14.65)	40,856 (14.46)	25,748 (13.79)	243,328
Private	7784 (3.31)	10,256 (3.48)	8807 (3.13)	8621 (2.62)	8135 (2.88)	4637 (2.48)	48,240

**Table 2 tropicalmed-07-00151-t002:** Number of children hospitalized with LRTIs during the COVID-19 pandemic (2020) compared to the previous five years (2015–2019).

	2015–2019	2020	*p* Value
Total	1,423,509 (88.41)	186,651 (11.59)	
Principal diagnosis, N (%)			<0.001
Pneumonia (J09–J18)	876,557 (61.58)	120,281 (64.44)	
Bronchiolitis (J21)	149,253 (10.48)	16,037 (8.59)	
Bronchitis (J20)	397,699 (27.94)	50,333 (26.97)	
Sex, N (%)			0.786
Male	835,121 (58.67)	109,400 (58.61)	
Female	588,388 (41.33)	77,251 (41.39)	
Age group, N (%)			<0.001
<1 year	336,673 (23.65)	42,247 (22.63)	
1–<5 years	829,390 (58.26)	103,718 (55.57)	
5–<18 years	257,446 (18.09)	40,686 (21.80)	
Region, N (%)			<0.001
Bangkok	58,567 (4.11)	6371 (3.41)	
Central	279,051 (19.60)	34,669 (18.57)	
East	81,594 (5.73)	8922 (4.78)	
Northeast	524,071 (36.82)	68,684 (36.80)	
North	122,316 (8.59)	18,428 (9.87)	
West	76,616 (5.38)	10,162 (5.44)	
South	281,294 (19.76)	39,415 (21.12)	
Hospital level, N (%)			<0.001
Primary	105,259 (7.39)	14,049 (7.53)	
Secondary	1,057,067 (74.26)	142,217 (76.19)	
Tertiary	217,580 (15.28)	25,748 (13.79)	
Private	43,603 (3.06)	4637 (2.48)	

**Table 3 tropicalmed-07-00151-t003:** Incidence per 1000 person-years during the COVID-19 pandemic (2020) compared to the previous five years (2015–2019).

Principal Diagnosis	2015–2019	2020	IRR (95%CI)
Pneumonia (J09–J18)	12.77	9.16	0.72 (0.71, 0.72)
Bronchitis (J20)	5.79	3.84	0.66 (0.65, 0.67)
Bronchiolitis (J21)	2.17	1.22	0.56 (0.55, 0.57)

IRR, incidence rate ratio; CI, confidence interval.

## Data Availability

The datasets generated and/or analyzed during the current study are not publicly available but are available from the corresponding authors (P.S., R.U.) upon request.

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
