# Peer review of "Effect of the COVID-19 Pandemic on Lower Respiratory Tract Infection Determinants in Thai Hospitalized Children: National Data Analysis 2015–2020"

_tropicalmed, 2022, doi:10.3390/tropicalmed7080151_

Round 1

Reviewer 1 Report

I read the manuscript with interest. It is well written, however; some English editions need to be done.

Obviously, social distancing, mask wearing, hygiene and lockdown will decrease the incidence of respiratory infection, in the other hand, the authors must outline other possible causes for lower diagnosis and hospitalization rate of respiratory tract infections, such as the population fear of admission during the pandemic, and the management/policy of health care providers to lower unnecessary admissions during the pandemic.

Moreover, the ultra long-term percussions to prevent COVID19 spreading, specially social distancing and lockdown, have some negative physical, emotional, economical and behavioural effects. Thus, these recommendations should be as short as possible. 

Author Response

Thank you for taking your time reviewing our manuscript. Your comments are very valuable for improving our writing. We are glad to receive all your valuable comments. Therefore, the authors have discussed, looked back, and edited the manuscript according to the constructive feedback of the manuscript. We really hope that our revision will match the criteria for publication in Tropical Medicine and Infectious Disease. Our responses to editors are described as follow.

Reviewer # 1

1. I read the manuscript with interest. It is well written, however; some English editions need to be done.

Response: Thank you very much for your feedback. We have already sent the texts to native language correction.

2. Obviously, social distancing, mask wearing, hygiene and lockdown will decrease the incidence of respiratory infection, in the other hand, the authors must outline other possible causes for lower diagnosis and hospitalization rate of respiratory tract infections, such as the population fear of admission during the pandemic, and the management/policy of health care providers to lower unnecessary admissions during the pandemic.

Response: Thank you very much for your feedback. In addition, "On the other hand, there may be other reasons for the lower hospitalization rate of LRTIs, such as people's fear of going to the hospital during the pandemic and health care pro-viders' efforts to cut down on admissions that aren't necessary during the pandemic." is added on the subject of discussion.

3. Moreover, the ultra long-term percussions to prevent COVID19 spreading, specially social distancing and lockdown, have some negative physical, emotional, economical and behavioural effects. Thus, these recommendations should be as short as possible.

Response: Thank you very much for your feedback. In addition, “however, social lockdown should be implemented as short as possible due to the economic and behavioral effects. [22]” is added on the subject of discussion.

Reviewer 2 Report

Thank you for the opportunity to review this manuscript. This manuscript is not ready for publication. There are some serious issues that the authors need to address. The following are my comments describing these issues:

1.                The title of the paper could benefit from the addition of the period of the study.

2.                The above suggestion should be implemented in the abstract and other parts of the manuscript if possible.

3.                The authors fail to justify in the Introduction the motivation to develop the study and the knowledge gap that they try to fill.

4.                The Introduction needs to include information on the potential impact of the measures implemented against COVID-19 on respiratory infections in general. For example, quarantine measures and the use of masks in the population influence the transmission of respiratory diseases in general.

5.                Authors should use the strobe guidelines to report their results.

6.                In the methods section, the absence of references to the information presented is observed.

7.                The authors should indicate the design of the study carried out.

8.                The methods section requires further expansion. For example, describe the place, and some characteristics of the study population.

9.                The authors do not explain how they selected the variables that were part of the study. Was it the product of a literature review?

10.             Variables should be clearly defined in the materials and methods section. For example, the level of the hospital is mentioned and it would be important to know what corresponds to primary, secondary, tertiary and private.

11.             Do the authors have an explanation for the high number of hospitalizations in 2018?

12.             Do the authors have an explanation for considering that the time of COVID-19 began in September 2019, as indicated in Figure 2?

13.             Likewise, the authors in Table 2 change the time cut-off points and only consider the year 2020 as the year of the pandemic and not the cut-off points in Figure 2.

14.             The authors must specify in the materials and methods section the source of the population data to obtain the incidence reported in Table 3. Is the population data from the population affiliated with health services?

15.             Authors should specify the time period for the incidence estimate described in Table 3

16.             Please add a deeper international discussion of the subject.

17.             The authors do not include the limitations of the study.

Author Response

Reviewer # 2

Thank you for the opportunity to review this manuscript. This manuscript is not ready for publication. There are some serious issues that the authors need to address. The following are my comments describing these issues:

1. The title of the paper could benefit from the addition of the period of the study.

Response: Thank you very much for your feedback. The study period was added to the title as “The Effect of the COVID-19 Pandemic on Lower Respiratory Tract Infection Determinants in Thai Hospitalized Children: The National Data Analysis 2015 – 2020”.

2. The above suggestion should be implemented in the abstract and other parts of the manuscript if possible.

Response: Thank you very much for your feedback. The study period was added to the title as shown in the manuscript line 20-21 and row 80.

3. The authors fail to justify in the Introduction the motivation to develop the study and the knowledge gap they try to fill.

Response: Thank you very much for your feedback; we revised the introduction to encourage further research and to fill knowledge gaps, as seen in row71-79.

4. The Introduction needs to include information on the potential impact of the measures implemented against COVID-19 on respiratory infections. For example, quarantine measures and the use of masks in the population influence the transmission of respiratory diseases in general.

Response: Thank you very much for your feedback; we revised the introduction to include information on the potential influence of COVID-19 prevention efforts on respiratory illnesses, as noted in rows 68-73.

5. Authors should use the strobe guidelines to report their results.

Response: Thank you very much for your feedback; we use the strobe guidelines to report our results as recommended by the reviewer.

6. In the methods section, the absence of references to the information presented is observed.

Response: Thank you very much for your feedback; we revised the manuscript and rechecked all references as recommended by the reviewer.

7. The authors should indicate the design of the study carried out.

Response: Thank you very much for your feedback; we revised the manuscript and added the study's design as recommended by the reviewer, as shown in row 86.

8. The methods section requires further expansion. For example, describe the place, and some characteristics of the study population.

Response: Thank you very much for your feedback; we revised the manuscript and added more details as recommended by the reviewer, as shown in rows 86-96.

9. The authors do not explain how they selected the variables that were part of the study. Was it the product of a literature review?

Response: Thank you very much for your feedback; we revised the manuscript and clarified the information in methods recommended by the reviewer, as shown in rows 101-111.

10. Variables should be clearly defined in the materials and methods section. For example, the level of the hospital is mentioned and it would be important to know what corresponds to primary, secondary, tertiary, and private.

Response: Thank you very much for your feedback; we revised the manuscript and clarified the information in methods recommended by the reviewer, as shown in rows 89-90.

11. Do the authors have an explanation for the high number of hospitalizations in 2018?

Response: Thank you for your comments; we amended the paper and added the high number of hospitalizations in 2018 as suggested by the reviewer, as displayed in row 164. However, we cannot explain the increased frequency of hospitalizations and must accept this as a limitation of our study.

12. Do the authors have an explanation for considering that the time of COVID-19 began in September 2019, as indicated in Figure 2?

Response: Thank you very much for your comments; we revised the paper and added the explanation of the COVID-19 pandemic in Thailand in part of the introduction, and we use the data for fiscal 2019 vs. 2020 to compare month to month, as shown in Figure 2.

13. Likewise, the authors in Table 2 change the time cut-off points and only consider the year 2020 as the year of the pandemic and not the cut-off points in Figure 2.

Response: Thank you very much for your comments; we revised the paper and added an explanation of the COVID-19 pandemic in Thailand in the introduction, and we use data from fiscal 2019 vs. 2020 to compare month to month, as shown in Figure 2, and we also compare the large set of data from fiscal years 2015-2019 vs. the fiscal year 2020 to see the trend of decreasing hospitalization, and we add more explanation to the result, as suggested by the reviewer.

14. The authors must specify in the materials and methods section the source of the population data to obtain the incidence reported in Table 3. Is the population data from the population affiliated with health services?

Response: Thank you very much for your comments; we revised the paper and added the source of the population data in row 82, as suggested by the reviewer.

15. Authors should specify the time period for the incidence estimate described in Table 3

Response: Thank you very much for your comments; we revised the paper and added a time period for the incidence estimate described in Table 3, as shown in rows 155-156, as suggested by the reviewer.

16. Please add a deeper international discussion of the subject.

Response: Thank you very much for your comments; we revised the manuscript and added more discussion, as shown in rows 168-212, as suggested by the reviewer.

17. The authors do not include the limitations of the study.

Response: Thank you very much for your comments; we revised the manuscript and added limitations, as shown in rows 168-212, as suggested by the reviewer.

Reviewer 3 Report

This study aims to identify and compare the incidence of lower respiratory tract infections among children in Thailand before and after introducing the COVID-19 virus. The study is retrospective and presents interesting findings. The authors concluded that NPIs may decrease the number of pediatric hospitalizations related to lower respiratory tract infections. The authors should pay attention to the text formatting. The authors should check authors guidelines prior to the submission of the revised version. 

The authors should add a statistical analysis sub-section. More details should be added to the statistical analysis sub-section.

Line 223: the authors must use the abbreviation instead of lower respiratory tract infections

Author Response

Thank you for taking the time to review our manuscript. Your comments are very valuable for improving our writing. We are glad to receive all your valuable comments. Therefore, the authors have discussed, looked back, and edited the manuscript according to the constructive feedback of the manuscript. We really hope that our revision will match the criteria for publication in Tropical Medicine and Infectious Disease. Our responses to editors are described as follows.

# Reviewer 3

This study aims to identify and compare the incidence of lower respiratory tract infections among children in Thailand before and after introducing the COVID-19 virus. The study is retrospective and presents interesting findings. The authors concluded that NPIs may decrease the number of pediatric hospitalizations related to lower respiratory tract infections.

1. The authors should pay attention to the text formatting.

Response: Thank you very much for your comments; we revised the manuscript and recheck the text formatting in the manuscript as recommended by the reviewer.

2. The authors should check authors guidelines prior to the submission of the revised version.

Response: Thank you very much for your comments; we revised the manuscript and recheck authors' guidelines prior to the submission of the revised version.

3. The authors should add a statistical analysis sub-section. More details should be added to the statistical analysis sub-section.

Response: Thank you very much for your comments; we revised the manuscript and add more details to the statistical analysis sub-section as shown in rows 94-104.

4. Line 223: the authors must use the abbreviation instead of lower respiratory tract infections

Response: Thank you very much for your comments; we revised the manuscript and use the abbreviation LRTIs instead of lower respiratory tract infections.

Reviewer 4 Report

The authors present a retrospective desk-based study on data from the Thailand Registry and correlate the averages used during the COVID-19 pandemic with the decrease in other lower respiratory tract infections in children. The introduction is well sized and the material and methods fit the objectives well, but they need to answer a question in the limitations section: 

Why could the decrease in respiratory infections not also be related to reduced accessibility to hospitals during the pandemic or under-diagnosis of respiratory infections?

Would it be possible to know mortality data for lower respiratory tract infections?

I believe that these questions need to be resolved in order to give more internal validity to the results provided.

A decrease in hosptialisations is not always good news and in this case should be very well justified by the authors as something positive.

Author Response

Thank you for taking the time to review our manuscript. Your comments are very valuable for improving our writing. We are glad to receive all your valuable comments. Therefore, the authors have discussed, looked back, and edited the manuscript according to the constructive feedback of the manuscript. We really hope that our revision will match the criteria for publication in Tropical Medicine and Infectious Disease. Our responses to editors are described as follows.

# Reviewer 4

The authors present a retrospective desk-based study on data from the Thailand Registry and correlate the averages used during the COVID-19 pandemic with the decrease in other lower respiratory tract infections in children. The introduction is well sized and the material and methods fit the objectives well, but they need to answer a question in the limitations section:

1. Why could the decrease in respiratory infections not also be related to reduced accessibility to hospitals during the pandemic or under-diagnosis of respiratory infections?

Response: Thank you very much for your feedback. In addition, "On the other hand, there may be other reasons for the lower hospitalization rate of LRTIs, such as people's fear of going to the hospital during the pandemic and health care providers' efforts to cut down on admissions that aren't necessary during the pandemic." is added on the subject of discussion.

2. Would it be possible to know mortality data for lower respiratory tract infections?

Response: Thank you very much for your comments; we are unable to remark on the mortality rate of this LRTI hospitalization because we lack the death rate of LRTIs in our data collection so we revised the manuscript and must accept this as a limitation of our study.

Round 2

Reviewer 1 Report

The authors made the suggested corrections

English editing still need to be done.

Author Response

Thank you for examining our manuscript thoroughly. Your feedback is invaluable for enhancing our writing. We are pleased to hear all of your insightful comments, and we apologize for the incomplete revision of the first round, which caused us to overlook certain crucial and significant points that you have generously brought to our attention. Accordingly, the authors have discussed, reviewed, and revised the work based on the constructive criticism received. We really hope that our amendment will meet Tropical Medicine and Infectious Disease's publication requirements. Our responses to editors are as outlined below.

Reviewer # 1

The authors made the suggested corrections

English editing still need to be done.

Response: Thank you very much for your valuable suggestion. This revision we put out for the expert English editing service of MDPI as the certification attached.

Reviewer 2 Report

-In general, the authors have not responded to the comments raised. I suggest that in the response letter you add and point out what changes you have made by copying and pasting the changed/corrected text.

-The authors do not explain what each of the mentioned levels corresponds to. Are they rural hospitals, are they specialized institutes?

-Even the authors do not adequately answer the question, Do the authors have an explanation for considering that the time of COVID-19 began in September 2019, as indicated in Figure 2? The COVID-19 pandemic did not start in September 2019.

-It is necessary to expand and provide greater details in the methodology so that the study can be replicated. It currently lacks details that do not make it replicable.

Author Response

Dear Reviewer

Thank you for examining our manuscript thoroughly. Your feedback is invaluable for enhancing our writing. We are pleased to hear all of your insightful comments, and we apologize for the incomplete revision of the first round, which caused us to overlook certain crucial and significant points that you have generously brought to our attention. Accordingly, the authors have discussed, reviewed, and revised the work based on the constructive criticism received. We hope our amendment will meet Tropical Medicine and Infectious Disease's publication requirements. Our responses to editors are as outlined below.

Reviewer # 2

  1. In general, the authors have not responded to the comments raised. I suggest that in the response letter you add and point out what changes you have made by copying and pasting the changed/corrected text.

Response: Thank you very much for your extremely valuable contributions to the improvement of our work. We sincerely apologize for overlooking any of your suggestions during the initial round of revisions; as a result, we will do our utmost to enhance our work in accordance with the details provided below.

1.1       The title of the paper could benefit from the addition of the period of the study.

Response: Thank you very much for your feedback. The study period was added to the title as “Effect of the COVID-19 Pandemic on Lower Respiratory Tract Infection Determinants in Thai Hospitalized Children: National Data Analysis 2015–2020”.

1.2       The above suggestion should be implemented in the abstract and other parts of the manuscript if possible.

Response: Thank you very much for your feedback. The study period was added to the abstract as shown in the manuscript as “A retrospective study was carried out in hospitalized children under the age of 18 in Thailand from October 2015 to September 2020.” and in the introduction as “Our study aimed to characterize the number of hospital admissions due to LRTIs in Thai children throughout the fiscal years 2015 and 2020”

1.3       The authors fail to justify in the Introduction the motivation to develop the study and the knowledge gap that they try to fill.

Response: Thank you very much for your feedback; we revised the introduction to initiate this study and to fill knowledge gaps, as “Since December 2019, the COVID-19 outbreak has not only had a devastating effect on public health, but also affected social and economic expenses. It has caused significant changes in human behavior, including social isolation, working from home, school and childcare facility closures, strict hygiene procedures, and widespread usage of face masks. It has been established that preventative hygiene activities such as hand washing can reduce the transmission of respiratory viruses. The majority of respiratory viruses are spread between individuals by droplet transmission [11]. Therefore, COVID-19 prevention strategies and alterations in human behavior may reduce the transmission of other respiratory viruses, which are a common cause of LRTIs in children [12]. However, the impact of COVID-19 preventive initiatives on childhood LRTIs is still uncertain. We expected NPIs to be able to minimize LRTIs, despite the fact that there is considerable doubt that the NPIs deployed during the outbreak reduced the incidence of certain LRTIs. Therefore, this study investigates the number of LRTIs in the years preceding and following the COVID pandemic.”

1.4       The Introduction needs to include information on the potential impact of the measures implemented against COVID-19 on respiratory infections in general. For example, quarantine measures and the use of masks in the population influence the transmission of respiratory diseases in general.

Response: Thank you very much for your feedback; we revised the introduction to include information on the potential influence of COVID-19 prevention efforts on respiratory illnesses, as “It has been established that preventative hygiene activities such as hand washing can reduce the transmission of respiratory viruses. The majority of respiratory viruses are spread between individuals by droplet transmission [11]. Therefore, COVID-19 prevention strategies and alterations in human behavior may reduce the transmission of other respiratory viruses, which are a common cause of LRTIs in children [12].”

1.5       Authors should use the strobe guidelines to report their results.

Response: Thank you very much for your feedback; we use the strobe guidelines to report our results as recommended by the reviewer.

1.6       In the methods section, the absence of references to the information presented is observed.

Response: Thank you very much for your feedback; we revised the manuscript and recheck all references as recommended by the reviewer and added reference to the information presented as “Patient diagnoses were identified using the International Statistical Classification of Diseases and Related Health Problems, 10th Revision, Thai Modification (ICD-10-TM) [13].”

1.7       The authors should indicate the design of the study carried out.

Response: Thank you very much for your feedback; we revised the manuscript and add the design of the study as recommended by the reviewer as “A retrospective study was carried out among Thai children hospitalized due to LRTIs under 18 years of age from October 2015 to September 2020.”

1.8       The methods section requires further expansion. For example, describe the place, and some characteristics of the study population.

Response: Thank you very much for your feedback; we revised the manuscript and add more details as recommended by the reviewer as “A retrospective study was carried out among Thai children hospitalized due to LRTIs under 18 years of age from October 2015 to September 2020. The periods within this study were divided according to Thailand’s fiscal year, which is from 1st October to 30th September each year. The data were extracted from the National Health Security Office database based on the Universal Coverage Health Security Scheme. Patient diagnoses were identified using the International Statistical Classification of Diseases and Related Health Problems, 10th Revision, Thai Modification (ICD-10-TM) [13]. The LRTIs were defined using ICD-10-TM for acute lower respiratory infection (J09–J21): pneumonia (J09–J18), bronchiolitis (J21), and bronchitis (J20).

We collected data on patients’ age, gender, month and year of admission, hospital level, hospital region, and hospital costs. Hospital level in Thailand was classified into four levels: primary care, secondary care, tertiary care, and private hospital. Primary care is typically located in rural areas, secondary care is typically located far from major cities and offers limited specialization, while tertiary care and private hospitals represent the highest degree of specialty treatment.”

1.9       The authors do not explain how they selected the variables that were part of the study. Was it the product of a literature review?

Response: Thank you very much for your feedback; we used the variables from the previous epidemiological study in Thailand reference [1] as the literature review and we revised the manuscript and clarify the information in methods as recommended by reviewer as “We presented the number of admissions due to LRTIs each year as monthly trends. As COVID-19 measures began in Thailand in the middle of March 2020, the dataset of admission numbers for the 2020 fiscal year (October 2019 to September 2020) was used to compare with the 2019 fiscal year (October 2018 to September 2019). Hence, the admission numbers from March to September 2020 represent post-COVID-19 data. The generalized estimating equation (GEE) was used to produce regression analysis of the number of admissions compared to the monthly trend of admission between the fiscal years 2019 and 2020. Because the number of admissions was the repeated measure variable (some patients may be hospitalized due to LRTIs many times during the study), the incidence of LRTIs was presented per 1000 person-years and divided into pneumonia, bronchiolitis, and bronchitis. [1] We used the incidence rate ratio to compare the incidence rate of each diagnosis between 2015–2019 and 2020.”

1.10    Variables should be clearly defined in the materials and methods section. For example, the level of the hospital is mentioned and it would be important to know what corresponds to primary, secondary, tertiary and private.

Response: Thank you very much for your comment. We add more explanation text as “Hospital level in Thailand was classified into four levels: primary care, secondary care, tertiary care, and private hospital. Primary care is typically located in rural areas, secondary care is typically located far from major cities and offers limited specialization, while tertiary care and private hospitals represent the highest degree of specialty treatment.”

1.11    Do the authors have an explanation for the high number of hospitalizations in 2018?

Response: Thank you for your comments; we amended the paper and added the high number of hospitalizations in 2018 as suggested by the reviewer. However, we are unable to explain the increased frequency of hospitalizations and must accept this as a limitation of our study, as mentioned in the limitation as “The final limitation is a lack of organism-specific data, which means that the etiology of LRTIs cannot be addressed; therefore, we cannot explain why the hospitalization rate in fiscal year 2018 is highest compared to previous fiscal years.”

1.12    Do the authors have an explanation for considering that the time of COVID-19 began in September 2019, as indicated in Figure 2?

Response: I greatly appreciate your comment. We amended “Figure 2” and corrected the information to be more comprehensible by adding "Year before the COVID-19 outbreak and Year of COVID-19" and using an arrow to highlight the timing of the COVID-19 pandemic.

1.13    Likewise, the authors in Table 2 change the time cut-off points and only consider the year 2020 as the year of the pandemic and not the cut-off points in Figure 2.

Response: I appreciate your feedback very much. Thank you so much for bringing out the important detail to us. We amended “Figure 2” and corrected the information to be more comprehensible by adding "Year before the COVID-19 outbreak and Year of COVID-19" and using an arrow to highlight the timing of the COVID-19 pandemic. So that it would be more comprehensible to the reader.

1.14    The authors must specify in the materials and methods section the source of the population data to obtain the incidence reported in Table 3. Is the population data from the population affiliated with health services?

Response: Thank you very much for your comments; we revised the paper and add the source of the population data in material and methods as “The data were extracted from the National Health Security Office database based on the Universal Coverage Health Security Scheme which is the main healthcare services in Thailand. Patient diagnoses were identified using the International Statistical Classification of Diseases and Related Health Problems, 10th Revision, Thai Modification (ICD-10-TM) [13]. The LRTIs were defined using ICD-10-TM for acute lower respiratory infection (J09–J21): pneumonia (J09–J18), bronchiolitis (J21), and bronchitis (J20).”

1.15    Authors should specify the time period for the incidence estimate described in Table 3

Response: Thank you very much for your comments; we revised the paper and added the time period for the incidence estimate described in Table 3 as “During the COVID-19 outbreak in Thailand in the 2020 fiscal year, a total of 186,651 pediatric patients were hospitalized with lower respiratory tract infections (LRTIs). There were 120,281 pneumonia admissions (64.44%), followed by 50,333 bronchitis admissions (26.97%) and 16,037 bronchiolitis admissions (8.59%). Comparing the data to the preceding five years as a benchmark, in the 2020 fiscal year, the incidence of lower respiratory tract infections among pediatric patients decreased statistically significantly (p value < 0.001; Table 2) throughout the COVID-19 period compared to the previous fiscal years. The incidence rates of pneumonia, bronchitis, and bronchiolitis were 9.16, 3.84, and 1.22 per 1000 people, respectively. The incidences were lower pre- and post-COVID-19 in all three diagnostic groups, with a 28% reduction in the pneumonia group (incidence rate ratio (RR) = 0.72; 95% confidence interval (CI): 0.71 to 0.72), a 44% reduction in the bronchiolitis group (IRR = 0.56; 95% CI: 0.55 to 0.57) and a 34% reduction in the bronchitis group (IRR = 0.66; 95% CI: 0.65 to 0.67) (Table 3).” as suggested by reviewer.

1.16    Please add a deeper international discussion of the subject.

Response: Thank you very much for your comments; we revised the manuscript and add more discussion as “After the COVID-19 epidemic, the number of pediatric outpatient clinics with respiratory tract infections fell concurrently with prior research in China [4]. After implementing community mitigation measures in response to the COVID-19 pandemic, prospective multicenter research conducted in seven U.S. communities revealed a drop in cases of acute respiratory tract infections in children [20]. During the statewide lockdown in Finland in 2020, the daily median rate of pediatric ER visits reduced considerably compared to the study period preceding the lockdown. Additionally, the influenza season was shorter and the weekly rate of new admission cases declined more quickly. [21]

After the containment of COVID-19 transmission, the lockdown ended in late May 2020, and the number of children admitted due to LRTIs remained lower than that in the previous year. Previous studies showed that parents were hesitant to bring their children to healthcare facilities during this pandemic out of concern about SARS-CoV-2 exposure. Hence, COVID-19’s influence on health care services may be to blame for the drop in hospitalizations, particularly among pediatric patients [22,23].” as suggested by the reviewer.

1.17    The authors do not include the limitations of the study.

Response: Thank you very much for your comments; we revised the manuscript and add limitations as “There are some limitations to the study. The main limitation is that these findings cannot be extrapolated to other countries. We gathered information on LRTIs that occurred in Thailand for the purposes of this study. In different nations, the COVID-19 pandemic has had varying degrees of impact. Every country has its own culture, policies, and management techniques that are exclusive to that country. Second, the data that we used came from a retrospective study known as the Universal Coverage Health Security Scheme Database. Furthermore, variations in data criteria could have created selection and reporting bias; thus, we are unable to remark on the mortality rate of LRTI hospitalization because we lack the death rate of LRTIs in our data collection. The final limitation is a lack of organism-specific data, which means that the etiology of LRTIs cannot be addressed; therefore, we cannot explain why the hospitalization rate in fiscal year 2018 is highest compared to previous fiscal years. This study, on the other hand, sheds light on a major shift in epidemiological trends caused by the present COVID-19 infection prevention strategies. The etiological pathogens of LRTIs may be the focus of future research because some pathogens may have an effect on these NPIs, as discovered in a number of countries in a previous study [26].” as suggested by reviewer.

  1. The authors do not explain what each of the mentioned levels corresponds to. Are they rural hospitals, are they specialized institutes?

Response: Thank you very much for your comment. We add more explanation text as “Hospital level in Thailand was classified into four levels: primary care, secondary care, tertiary care, and private hospital. Primary care is typically located in rural areas, secondary care is typically located far from major cities and offers limited specialization, while tertiary care and private hospitals represent the highest degree of specialty treatment.”

  1. Even the authors do not adequately answer the question, Do the authors have an explanation for considering that the time of COVID-19 began in September 2019, as indicated in Figure 2? The COVID-19 pandemic did not start in September 2019.

Response: I greatly appreciate your comment. We amended Figure 2 and corrected the information to be more comprehensible by adding "Year before the COVID-19 outbreak and Year of COVID-19" and using an arrow to highlight the timing of the COVID-19 pandemic.

  1. It is necessary to expand and provide greater details in the methodology so that the study can be replicated. It currently lacks details that do not make it replicable.

Response: Many thanks for your insightful feedback. We updated our entire process such that it could be replicated with the dataset we utilized. Our study analyzes the ICD-10-classified nationwide dataset mentioned in the methodology. This is the revised methodology “A retrospective study was carried out among Thai children hospitalized due to LRTIs under 18 years of age from October 2015 to September 2020. The periods within this study were divided according to Thailand’s fiscal year, which is from 1st October to 30th September each year. The data were extracted from the National Health Security Office database based on the Universal Coverage Health Security Scheme which is the main healthcare service in Thailand. Patient diagnoses were identified using the International Statistical Classification of Diseases and Related Health Problems, 10th Revision, Thai Modification (ICD-10-TM) [13]. The LRTIs were defined using ICD-10-TM for acute lower respiratory infection (J09–J21): pneumonia (J09–J18), bronchiolitis (J21), and bronchitis (J20).

We collected data on patients’ age, gender, month and year of admission, hospital level, hospital region, and hospital costs. Hospital level in Thailand was classified into four levels: primary care, secondary care, tertiary care, and private hospital. Primary care is typically located in rural areas, secondary care is typically located far from major cities and offers limited specialization, while tertiary care and private hospitals represent the highest degree of specialty treatment. The institutional review board of Khon Kaen University approved this study (#HE641151).

All statistical analyses were performed using STATA software version 10 (StataCorp LP). Categorical data were described using frequencies and percentages. Continuous data are expressed as the mean and standard deviation. We presented the number of admissions due to LRTIs each year as monthly trends. As COVID-19 measures began in Thailand in the middle of March 2020, the dataset of admission numbers for the 2020 fiscal year (October 2019 to September 2020) was used to compare with the 2019 fiscal year (October 2018 to September 2019). Hence, the admission numbers from March to September 2020 represent post-COVID-19 data. The generalized estimating equation (GEE) was used to produce regression analysis of the number of admissions compared to the monthly trend of admission between the fiscal years 2019 and 2020. Because the number of admissions was the repeated measure variable (some patients may be hospitalized due to LRTIs many times during the study), the incidence of LRTIs was presented per 1000 person-years and divided into pneumonia, bronchiolitis, and bronchitis. [1]   We used the incidence rate ratio to compare the incidence rate of each diagnosis between 2015–2019 and 2020. The 95% confidence interval (CI) of the rate was computed based on the normal approximation to the binomial distribution. P < 0.05 was considered to indicate statistical significance.”
